# Upper Limb Function Recovery by Combined Repetitive Transcranial Magnetic Stimulation and Occupational Therapy in Patients with Chronic Stroke According to Paralysis Severity

**DOI:** 10.3390/brainsci13020284

**Published:** 2023-02-08

**Authors:** Daigo Sakamoto, Toyohiro Hamaguchi, Kai Murata, Hiroshi Ito, Yasuhide Nakayama, Masahiro Abo

**Affiliations:** 1Department of Rehabilitation Medicine, The Jikei University School of Medicine Hospital, Tokyo 105-8471, Japan; 2Department of Rehabilitation, Graduate School of Health Science, Saitama Prefectural University, Saitama 343-8540, Japan; 3Department of Rehabilitation Medicine, The Jikei University School of Medicine, Tokyo 105-8461, Japan

**Keywords:** stroke, occupational therapy, activities of daily living, goal-setting, transcranial magnetic stimulation, upper extremity, motor paralysis, neurorehabilitation

## Abstract

Repetitive transcranial magnetic stimulation (rTMS) with intensive occupational therapy improves upper limb motor paralysis and activities of daily living after stroke; however, the degree of improvement according to paralysis severity remains unverified. Target activities of daily living using upper limb functions can be established by predicting the amount of change after treatment for each paralysis severity level to further aid practice planning. We estimated post-treatment score changes for each severity level of motor paralysis (no, poor, limited, notable, and full), stratified according to Action Research Arm Test (ARAT) scores before combined rTMS and intensive occupational therapy. Motor paralysis severity was the fixed factor for the analysis of covariance; the delta (post-pre) of the scores was the dependent variable. Ordinal logistic regression analysis was used to compare changes in ARAT subscores according to paralysis severity before treatment. We implemented a longitudinal, prospective, interventional, uncontrolled, and multicenter cohort design and analyzed a dataset of 907 patients with stroke hemiplegia. The largest treatment-related changes were observed in the Limited recovery group for upper limb motor paralysis and the Full recovery group for quality-of-life activities using the paralyzed upper limb. These results will help predict treatment effects and determine exercises and goal movements for occupational therapy after rTMS.

## 1. Introduction

Motor paralysis after stroke limits patients’ activities of daily living (ADL) and reduces their quality of life [1,2]. Recently, noninvasive brain stimulation therapy has been developed to improve patients’ motor paralysis and ADL, and its effectiveness has been demonstrated [3,4]. The treatment of upper limb motor paralysis involves modulation of interhemispheric inhibition and induction of neuroplasticity in the cerebrum. A novel intervention using repetitive transcranial magnetic stimulation (rTMS) in combination with intensive occupational therapy (NEURO) has recently been developed [5]. In patients with stroke hemiplegia, high-frequency rTMS has been applied to the hemisphere ipsilateral to the paralysis to increase excitability [6], and low-frequency rTMS has been applied to the contralateral hemisphere to decrease interhemispheric inhibitory connections [7,8] with the damaged cortex [9]; thus, both high-frequency rTMS and low-frequency rTMS have been applied [10]. Repetitive currents are induced in the brain cortex to produce long-term changes in cortical excitability. In acute patients, high-frequency (10 Hz) rTMS applied to the impaired motor cortex activates it, improving paralysis [11,12]. In occupational therapy after rTMS, the patients in whom the activation of the interhemispheric inhibitory motor cortex has been adjusted are prescribed repetitive joint movements. The aim is to promote use-dependent plasticity in the brain and to subsequently restore motor paralysis and improve ADL [13]. NEURO is an effective treatment for improving upper limb dysfunction and impairments in ADL in chronic stroke patients 6 months after stroke onset. Its therapeutic effect has been shown to be unaffected by stroke type (cerebral hemorrhage or cerebral infarction) [14].

The goal of NEURO is to improve the quality of movement of the patient’s paralyzed upper limb by allowing it to be used in ADL. Since the effectiveness of NEURO depends on the severity of motor paralysis, therapists determine the exercises and target movements based on the patient’s pre-treatment upper limb function assessment score. The Fugl–Meyer Assessment of the Upper Extremity (FMAUE) and the Action Research Arm Test (ARAT) are used to assess upper limb motor function outcomes in NEURO [15]. These evaluation methods have been shown to have high accuracy and clinical usefulness. A previous study has been conducted to estimate post-treatment scores from the pre-NEURO FMAUE score [16]. The ARAT is a functional upper limb assessment tool used in patients with post-stroke hemiplegia and is characterized by its ability to reflect the patient’s activity [17]. Since the ARAT consists of object manipulation and reaching tasks, the occupational therapist (OT) plans exercises by estimating the ADLs in which the patient can use their hands based on the obtained assessment results. As the ARAT score correlates with the Motor Activity Log, which investigates the use of the paralyzed limb in ADLs, OTs helping patients improve their activity limitations can use it as a reference value for exercises and goal-setting [18,19]. Therefore, it can be inferred that predicting treatment effects with ARAT is more advantageous than using FMAUE in setting treatment goals and planning effective ADL exercises for patients. If ARAT scores are found to improve with NEURO, it will be easier for OTs to pre-determine the content of ADL exercises and develop achievable ADL goals.

Patients with mild-to-moderate motor paralysis with FMAUE scores ≥43 have higher interhemispheric inhibition from the healthy hemisphere to the affected hemisphere. It is predicted that the therapeutic effect of upper limb practice in the presence of rTMS-induced changes in synaptic transmission efficiency is dependent on motor paralysis severity [20]. If the post-treatment effects according to motor paralysis severity can be predicted using pre-treatment ARAT scores, the target movements for patients could be set with high accuracy. Recently, a treatment method using a brain-computer interface (BCI) was developed for the rehabilitation of stroke patients, and its effectiveness has been reported [21,22]. Even for new intervention methods, it is better to formulate exercises adapted to the severity of paralysis and recovery. Therefore, the results obtained in this study can be used as data to plan the most appropriate practice for patients in terms of future new intervention methods. As a result, this study aimed to estimate the amount of change in ARAT scores for each level of motor paralysis severity, classified according to the ARAT score before NEURO.

## 2. Materials and Methods

### 2.1. Study Design

In this multicenter, longitudinal, prospective, interventional, uncontrolled study, we reviewed the medical records of patients with stroke from February 2017 to March 2021 from 6 different hospitals in Japan certified as NEURO implementation facilities. These included Izumi Memorial Hospital, Shimizu Hospital, Nishi-Hiroshima Rehabilitation Hospital, Tokyo General Hospital, Kyoto O’Hara Memorial Hospital, and Tokyo Jikei University Hospital. Izumi Memorial Hospital is located in Tokyo and is certified as a community rehabilitation support center. Shimizu Hospital is located in the Tottori Prefecture, about 900 km west of Tokyo, and mainly provides orthopedic surgery and rehabilitation medicine treatments. Nishi-Hiroshima Rehabilitation Hospital is located in the Hiroshima prefecture, about 800 km west of Tokyo, and specializes in rehabilitation for patients who have passed the acute stroke phase. Tokyo General Hospital is located in Tokyo and is a general hospital with 30 departments. Kyoto O’Hara Memorial Hospital is located in the Kyoto prefecture, about 500 km west of Tokyo, and is a specialized facility for rehabilitation. Tokyo Jikei University Hospital is located in Tokyo and is a university hospital that provides advanced medical treatment. We evaluated the therapeutic effects of NEURO by providing patients with selected functional exercises based on the severity of their motor paralysis. We also attempted to define a research protocol in this study.

### 2.2. Ethics Statements

Patients were not required to provide informed consent because the analysis used anonymous clinical data obtained after each patient had agreed to undergo NEURO by providing written consent. This study was approved by the Jikei University School of Medicine Ethics Committee (approval number 24-295-7061).

### 2.3. Participants

In this study, we selected patients who presented at an accredited facility due to wishing to undergo NEURO. Patients who received NEURO according to the rTMS guidelines during the study period, who were aged ≥18 years, had been diagnosed with stroke for >12 months and had no cognitive impairment (pre-treatment Mini-Mental State Examination score >26) were included [23]. The exclusion criteria were as follows: no ARAT score data, subarachnoid hemorrhage, arteriovenous malformation, brain tumor, diagnosis of childhood paralysis, and bilateral motor paralysis. 

### 2.4. Sample Size

The ARAT score was used as a quantitative variable, and variations in the ARAT scores among the five groups (see *Statistical analysis* for pre-treatment severity) were compared using G*power with analysis of covariance (ANCOVA; F-test, main effects) using the following values: effect size f = 0.25, α = 0.05, 1 − β = 0.80, number of groups = 5, and number of covariates = 5. Therefore, the total number of patients required was 242 (49 × 5 groups).

### 2.5. rTMS Combined with Occupational Therapy

NEURO was performed at an accredited facility in Japan, where physicians and therapists who had completed the prescribed training treated the patients according to the NEURO protocol. All patients were hospitalized for 15 days and received rTMS and occupational therapy [5]. Patients received a maximum of 6 sessions of occupational therapy per day, with each session lasting 20 min. Physiotherapy was occasionally prescribed for two to three of the 6 sessions, depending on the patient’s complaints and state of physical function. Physiotherapy mainly consisted of muscle stretching, stimulating exercises, balance exercises, ADL exercises, and gait exercises, with the goal of improving the patient’s ability to mainly walk, stand, and perform basic movements. The allocation of occupational and physical therapy sessions was determined by the physician in charge. 

Occupational therapy was conducted as one-to-one training with the goal of regaining the use of the paralyzed upper limb in daily life. The OT determined the target movements together with the patient, based on the patient’s wishes and the results of the physical function assessment. To achieve the goal, the OT prescribed the following to the patients: functional exercises for the proximal and distal parts of the upper limb, skillful movement exercises using objects, daily living exercises to use the paralyzed side, lifestyle guidance to promote the use of the paralyzed side, and self-guided exercises to improve motor paralysis. The exercises were individualized, taking into account the severity of the motor paralysis and the patient’s goals. However, the main treatment modalities were as follows: for patients with severe motor paralysis and severe spasticity, muscle stretching, joint mobilization exercises, and proximal upper extremity training were prioritized to allow the patient the use of the affected hand as an adjunct in ADLs. Patients with moderate motor paralysis were given exercises to promote isolated movement and real-life activities to restore motor function and improve ADLs. Patients with mild motor paralysis were given coordination and manipulation exercises to enable them to perform more challenging ADLs requiring arm-hand coordination [13,24]. Treatments used in conjunction with occupational therapy included practice with visual stimulation of mirror images, transcutaneous electrical nerve stimulation, repetitive peripheral sensory stimulation, and muscle-tendon vibration [25,26,27,28]. 

Patients received rTMS daily, excluding holidays [5,13,24]. A 70-mm figure-eight coil, attached to a MagPro R100 stimulator (MagVenture Company, Farum, Denmark), was used for rTMS during each session, with one of the following methods: (1) focal 1-Hz rTMS applied to the contralesional hemisphere over the primary motor area, as described by previous studies, (2) rTMS over the hand area of the ipsilesional primary motor cortex (M1) for a duration of 30 trains of 50 pulses with 25-s intervals at 10 Hz and 90% resting motor threshold (RMT) (total, 1500 pulses/day), (3) bilateral sequential stimulation involving low-frequency (1 Hz) contralesional stimulation followed immediately by high-frequency (10 Hz) stimulation in the ipsilesional primary motor cortex, or (4) theta-burst stimulation. The protocol consisted of bursts containing 3 pulses at 50 Hz, repeated at 5 Hz intervals (20 ms between each stimulus) but applied in 2-s trains repeated every 10 s for a total of 190 s (600 pulses in total). The attending physician defined the stimulation intensity as the lowest intensity, which was set to 90% of the RMT for the first dorsal interosseous muscle.

### 2.6. Outcomes

The change in ARAT scores was used to assess the primary outcome. The ARAT is an upper extremity function assessment developed based on the Upper Extremity Function Test [29]. The ARAT consists of 4 subtests: grasp, grip, pinch, and gross movement [30]. For the grasp subtest, a block, cricket ball, and grinding stone are used; for the grip subtest, a glass, cylinder, and washer are used; and for the pinch subtest, a metal ball and marble are used. In these subtests, the patient moves an object to a specified position or performs a movement with an object according to the instructions. In gross movement, patients reach toward the back of their head, the top of their head, and the mouth with their hands. The ARAT is scored on a 4-point ordinal scale, wherein: 0 = unable to perform any part of the test, 1 = able to perform test partially, 2 = able to complete the test but takes an abnormally long time or has great difficulty, and 3 = able to perform test normally. With reference to a 57-point scale, ARAT scores of 0–10, 11–21, 22–42, 43–54, and 55–57 are construed to represent no, poor, limited, notable, and full recovery capacity, respectively [31]. Muscle spasticity of the paretic arm was assessed using the modified Ashworth Scale (MAS).

Changes in Jikei Assessment Scale for Motor Impairment in Daily Living (JASMID) scores were used as the secondary outcome. The JASMID is a patient-reported measure for investigating paralyzed side upper-limb use in ADL among patients with stroke hemiplegia. It is an assessment measure that has previously been validated for reliability and validity [32]. A similar assessment is the Motor Activity Log; however, the JASMID includes questions adapted to the Japanese lifestyle [33]. Patients are asked to respond to each question on a 5-point quantitative scale (0 = never, 3 = sometimes, and 5 = always) to determine the frequency of paralyzed upper limb use and on another 5-point qualitative scale (0 = almost no use, 3 = experiencing moderate difficulty, and 5 = experiencing no difficulty at all) to determine ability to use of the upper limb). The quantity and quality scores are then calculated based on the scores of a total of 20 questions. 

### 2.7. Statistical Analysis

Patients’ ARAT scores were divided into five groups (no, poor, limited, notable, and full) according to pre-treatment severity and used as fixed factors in the ANCOVA. The severity of paralysis is a predictor of ARAT scores [31]. ARAT scores were converted from the total grasp, grip, pinch, and gross movement subscores into a delta value (post-treatment minus pre-treatment), which was used as a dependent variable in the ANCOVA. To estimate post-treatment recovery from pre-treatment paralysis severity, a multinomial logistic regression analysis was performed using delta values of ARAT and JASMID as dependent variables and pre-treatment motor paralysis severity as a predictor. Ordinal logistic regression analysis was used to compare changes in ARAT subscores in patients categorized according to the pre-treatment paralysis severity. 

Recovery of upper limb motor paralysis is affected by neuromodulation with rTMS and spasticity treatment [34]. Age, sex, side of paralysis, and time since onset were included as potential confounders when comparing the effects of NEURO between groups in a previous study [35]. Therefore, we also included these factors as confounders in the covariates of the ANCOVA and ordinal logistic regression analysis. JASP 0.16 (https://jasp-stats.org/ accessed on 1 August 2022) software was used for statistical analysis. A *p*-value of <0.05 was considered statistically significant.

## 3. Results

### 3.1. Participants

A total of 2022 patients with stroke who met the NEURO eligibility criteria were treated at the six hospitals. ARAT data were unavailable for 1096 patients, and 19 patients who met the exclusion criteria were excluded. Therefore, the final analysis included 907 patients (Figure 1). Variations in the total ARAT scores (mean ± standard deviation) before and after NEURO for each severity group were as follows: hospital A, 3.6 ± 4.2; hospital B, 1.7 ± 4.9; hospital C, 3.2 ± 4.0; hospital D, 2.8 ± 4.4; hospital E, 3.4 ± 5.1; and hospital F, 3.7 ± 5.1.

### 3.2. Descriptive Data

The patients’ clinical characteristics are presented in Table 1. Patients were categorized into the “No” recovery group, 275 patients (30%); “Poor” recovery group, 167 patients (18%); “Limited” recovery group, 269 patients (30%); “Notable” recovery group, 84 patients (9%), and “Full” recovery group, 112 patients (12%). rTMS included low-frequency stimulation on the intact brain hemispheres in 708 patients (77.9%), high-frequency rTMS on the lesion-side hemisphere in one patient (0.1%), and theta burst stimulation on the intact hemisphere in 198 patients (21.8%). Seven patients (0.8%) were treated with botulinum toxin A or xylocaine during NEURO.

### 3.3. Outcome Data

The patients’ clinical characteristics according to ARAT severity classifications are presented in Table 2. The total ARAT scores (median (25th, 75th percentile)) before treatment were 4 (3, 6) in the No recovery group, 16 (13, 18) in the Poor recovery group, 30 (26, 36) in the Limited recovery group, 48 (45, 51) in the Notable recovery group, and 57 (56, 57) in the Full recovery group. The JASMID quantity and quality scores were highest in the Full recovery and lowest in the No recovery groups. The results of changes in ARAT and JASMID scores for patients classified by ARAT severity are shown in Table 3.

### 3.4. Main Results

Delta values of the total ARAT scores before and after NEURO were analyzed according to the severity of paralysis using ANCOVA. The results are displayed in Table 4. The delta values (mean ± standard deviation) were 2.4 ± 4.2 in the No recovery group, 3.9 ± 4.8 in the Poor recovery group, 4.6 ± 5.8 in the Limited recovery group, 3.0 ± 4.2 in the Notable recovery group, and 0.3 ± 1.3 in the Full recovery group (Table 3), indicating a significant main effect by group (F = 18.677, *p* < 0.001, η^2^ = 0.077) age, sex, laterality in the paretic and dominant hands, and time since onset were adjusted for as covariates. This main effect remained unchanged when adjusted for the pre-treatment ARAT total score, indicating a significant main effect (F = 18.545, *p* < 0.001, η^2^ = 0.076). A post-test for the change in total ARAT scores showed that the Limited and Notable recovery groups displayed a more significant change in total ARAT scores than the Full recovery group (*p* = 0.024 and *p* = 0.022). Because muscle spasticity influenced upper extremity paraplegia recovery, the patients’ pretreatment data were reanalyzed with the main outcome as a covariate, and this did not affect the results (MAS_elbow_; F = 2.278, *p* = 0.132, η^2^ = 0.002).

In the ARAT subscore analysis, there was a significant main effect of the group on the delta values for subscores A–D (*p <* 0.05, adjusted for all covariates; Table 4). The post-test results showed that in terms of ARAT total scores, the changes in the Poor and the Limited recovery groups were significantly greater than that in the No recovery group (*p* = 0.034 and *p* < 0.001, respectively, Figure 2a). The change in the Notable recovery group was significantly greater than that in the Full recovery group (*p* = 0.003). Regarding the ARAT grasp score, the changes in the No, Poor, and Limited recovery groups were significantly greater than that in the Full recovery group (*p* = 0.005, *p* < 0.001, and *p* < 0.001, respectively, Figure 2b). Further, in terms of the ARAT grip score, the changes in the Poor and Limited recovery groups were significantly greater than that in the Full recovery group (both *p* < 0.001, respectively, Figure 2c). In terms of the ARAT pinch score, the change in the Limited recovery group was significantly greater than that in the No, Poor, and Full recovery groups (*p* < 0.001, *p* = 0.002, and *p* < 0.001, respectively, Figure 2d). The change in the Notable recovery group was significantly greater than that in the No and Full recovery groups (both *p* < 0.001). Regarding the ARAT gross movement scores, the changes in the No, Poor, and Limited recovery groups were significantly greater than that in the Full recovery group (*p* = 0.014, *p* = 0.012, and *p* = 0.004, respectively, Figure 2e). In terms of the post-test ARAT subscores, the change in the Notable recovery group was not significantly different from those in the Poor and Limited recovery groups. The JASMID score of hand usage (quantity, F = 2.02, *p* = 0.089, η^2^ = 0.009) had no main effect of the group on the delta value. On the other hand, the JASMID score of satisfaction did have a main effect of the group on the delta values (quality, F = 6.66, *p* < 0.001, η^2^ = 0.028, Table 4). According to the multiple comparisons test, the JASMID score of quality had a significantly greater main effect of group on the delta value in the Limited and Full recovery groups than in the No recovery group (*p* = 0.032 and *p* < 0.001, respectively, Figure 2f).

Next, stratified analysis was performed using multinomial regression to factor changes in ARAT and JASMID scores into the severity of motor paralysis prior to treatment. Akaike’s Information Criterion (AIC) = 2677, grasp (x^2^ = 36.2, *p* < 0.001, AIC = 2724), grip (x^2^ = 31.9, *p* < 0.001, AIC = 2728), pinch (x^2^ = 124, *p* < 0.001, AIC = 2635), gross movement (x^2^ = 21.1, *p* < 0.001, AIC = 2739), and JASMID quality (x^2^ = 24.8, *p <* 0.001, AIC = 2735) showed a significant fit to the model by the severity of motor paralysis, but none in JASMID quantity (x^2^ = 6.52, *p* = 0.16, AIC = 2753). Coefficients of variation and odds ratios were calculated for the ARAT total score, subscores A–D, and the change in JASMID quantity and quality scores with respect to the No recovery group data for each severity level (Figure 3).

Regarding total ΔARAT, when the coefficient of variation of the No recovery group was 1, that of the Poor recovery group was 0.9, that of the Limited recovery group was 1.1, and that of the Notable recovery group was 0.8; meanwhile, the odds ratio was estimated to be 2.53 times that of the No recovery group for the Poor recovery group, 3.08 times that of the No recovery group for the Limited recovery group, and 2.22 times that of the No recovery group for the Notable recovery group. The coefficient of variation for the Full recovery group was −0.9, and the odds ratio was 0.42. Regarding ΔJASMID, when the coefficient of variation of the No recovery group was 1, that of the Limited recovery group was 0.6, that of the Notable recovery group was 0.8, and that of the Full recovery group was 0.8; meanwhile, the odds ratio was estimated to be 1.84 times that of the No recovery group for the Limited recovery group, 2.30 times that of the No recovery group for the Notable recovery group, and 2.23 times that of the No recovery group for the Full recovery group (see Appendix A).

## 4. Discussion

In this study, the amount of change in ARAT was calculated for each level of motor paralysis severity, classified according to the ARAT score before NEURO. The results indicated that patients in the Limited recovery group experienced the most significant improvement in upper limb motor paralysis with NEURO. In a previous study, patients with FMAUE scores ≥43 had higher interhemispheric inhibition from the healthy hemisphere to the affected hemisphere [20]. In the present study, all the patients were irradiated with rTMS, and in 99% of them, the irradiation site was the primary motor cortex of the intact hemisphere. The higher interhemispheric inhibition in the Limited recovery group and the higher excitability of the primary and supplementary motor cortexes on the lesion side achieved with rTMS may have caused the greater upper limb motor paralysis caused by the subsequent occupational therapy [20,36].

In the present study, the degree of change in the frequency of hand use did not differ according to motor paralysis severity; however, the quality of movement was most greatly improved in the Full recovery group. A phenomenon termed “Learned non-use hand” that affects the paretic arm or hand occurs in patients with stroke paraplegia due to the continuous disuse of the paralyzed upper limb in their daily lives [37,38]. In the present study, the Full recovery group had the highest scores in the pre-treatment evaluation and had less functional impairment of the upper limb than the other groups. In the intensive inpatient rehabilitation program, in which OTs provided appropriate practice and instructions for performing ADLs, the degree of improvement in the frequency of hand use did not differ by severity; nevertheless, the Full recovery group was presumably more satisfied with the use of their hands and had greater qualitative improvement than the other groups. The effectiveness of rehabilitation is enhanced when patients themselves are aware of the motor functions they need to perform to carry out the ADL set as a goal by them and the therapist [39,40]. The results obtained in this study may be used as a reference for determining the content of occupational therapy exercises and ADL goals for use with rTMS; this data is summarized in Appendix B.

The No recovery group was predicted to have a more significant improvement in the ARAT grasp and gross movement scores by NEURO than the other groups. However, the quality of paralyzed upper limb movement was predicted to show a smaller improvement. Our results are consistent with the findings of previous studies reporting that patients with severe motor paralysis regained function in the proximal part of the upper limb [41,42]. Regarding NEURO occupational therapy, it can be inferred that ADL exercises that frequently use shoulder and elbow movements are suitable for patients with severe motor paralysis, as they improve the function of the proximal part of the upper limbs. In addition, it is recommended to practice movements with the paralyzed upper limb according to the patient’s wishes and to set up target movements to improve the quality of hand use. In the ARAT, the grasp task includes rotation of the forearm, and the gross movement task includes reaching the patient’s own body. For the target daily activities, it is suggested to use the proximal part of the upper limb to hold in place a plate, paper, or a book on a desk, a task that involves inward movement of the forearm. Similarly, even in cases of reduced hand dexterity, if the patient can reach their own body, they can aim to acquire movements such as lifting an upper garment when opening and closing a zipper, washing the upper limb and fingers of the non-paralyzed side, smoothing creases in clothes, and removing dust from clothes [43,44].

The Poor recovery group was more likely to show improvement in ARAT grasp and grip scores using NEURO. However, these patients were less likely to show improvement in ARAT pinch scores. It is assumed that improvement in hand function is necessary to improve the use of the paralyzed upper limb and the movement quality of the Poor recovery group patients. The ARAT grip task involves holding the forearm in the middle position or moving it from the medial to the external rotation position. Using occupational therapy, we expect that the grip score will improve. The target movements include grasping a plastic bottle, opening and closing a sliding door, holding a toothbrush while applying toothpaste, flipping a switch or pressing an elevator button within reach using the paralyzed limb, and grasping a cell phone with the paralyzed hand [45,46].

Since the Limited recovery group showed the most promising improvement in ARAT scores with NEURO, the target movement practices should be defined by estimating the target degree of improvement in grasp, grip, gross movement, and pinch. In rehabilitation, an improvement in motor function score does not always directly lead to an improvement in the amount of use or quality of movement possible with the paralyzed upper limb [47]. This may be because patients can perform compensatory movements with the paralyzed fingers and upper limbs and do not use the improved hand functions properly. Therefore, daily living exercises are essential in occupational therapy. Learned bad use of the paretic limb reinforces abnormal compensatory movement strategies at the expense of normal movement patterns, making it difficult to improve movement performance [48,49]. In occupational therapy, it is important to provide appropriate feedback for abnormal joint movements and inhibit learned bad use so that patients can appropriately use their paralyzed upper limbs in their ADLs. Although this point has not been tested, the patient’s target movements should be incorporated into the practice. The patient should be motivated to thoroughly practice using the paralyzed upper limb for ADLs so that they can perform the function when their motor function improves. The suggested target activities include manipulating a spoon or fork, zipping and unzipping clothes, buttoning and unbuttoning clothes, putting on socks, tying shoelaces, washing one’s face, writing one’s signature, and other activities requiring fine motor skills [50,51].

The Notable recovery group had fewer ARAT and JASMID subscores showing significant improvement compared to the other groups. Use-dependent plasticity is a phenomenon in which the same pattern of activity tends to occur when specific neurons are repeatedly activated; this is the aim of repetitive practice in rehabilitation [52,53]. In recent years, the practice time and the number of joint movements required to achieve recovery of motor function have been assessed [54,55]. In occupational therapy included in NEURO, it is important to promote use-dependent plasticity and improve motor function and ADL by providing a sufficient amount of challenging movement exercises for patients with Notable-level scores. For patients with Notable-level scores and good proximal and distal function, we propose the following ADLs as goals: drinking water from a cup, drying laundry, washing hair, tying hair, manipulating chopsticks, and tying a necktie [44,56,57].

In the Full recovery group, the change in the ARAT score was small, but the change in JASMID quality was expected to be high. The recovery of motor paralysis improved in terms of motor speed and coordination after patients regained joint movements, experienced weakening spasticity and gained the ability to perform isolated movements. Studies examining the difficulty level of detailed FMAUE items also support this recovery process [58,59,60]. A method using a motion analyzer to detect angular and velocity changes in joints is recommended for evaluating functional impairment and determining treatment effects in patients with mild hemiplegia [61]. The Box and Block Test and the Wolf motor function test can assess the coordination of movement and speed of joint movement based on the number of blocks carried and the time required to perform the task [62,63,64]. However, the ARAT is scored on an ordinal scale, and it is difficult to evaluate these factors in detail. In other words, the Full recovery group is expected to show improved quality of movement by improving the speed of movement and coordination of joint movements, which would be challenging to evaluate using the ARAT. Therefore, occupational therapy in NEURO is recommended to provide more challenging exercises, such as those for adjusting the speed of joint movements, for complex movements that require multiple joints, and for resistance exercises. Regarding ADL, it is possible to aim to acquire activities that require hand dexterity and upper limb motor coordination, such as cooking, brushing teeth, operating smartphones and PCs, and putting on and taking off necklaces and earrings [65,66]. In addition, this can be expected to improve the quality of ADLs to a level desired by the patient.

The results of this study can be applied to new interventions, including BCI, when planning exercises appropriate to the severity of the patient’s condition. For example, a treatment in which BCI was applied to exercise therapy provided by therapists was reported to result in greater improvement in patients’ motor paralysis [67,68]. Although the therapeutic effects of BCI have been demonstrated, a method to predict patient recovery and to set motor tasks appropriate for their severity has not been formulated [69]. The results of this study provided data to predict the amount of recovery after treatment for patients to whom newly developed interventions will be applied and to plan effective exercise tasks according to the severity of the patient’s illness. 

This study has several limitations. The ARAT used for the main outcome reportedly has a ceiling effect, and it is inferred that changes in patients with mild motor paralysis may be underestimated [70]. Sensory perception, muscle tone, and joint range of motion are involved in the ability to manipulate objects [71,72,73,74]. The present study does not clarify the effects of the presence or absence of these symptoms on the degree of change in ARAT scores. Since the JASMID questionnaire was designed for the Japanese lifestyle, it is unclear whether the same results would be obtained if it was administered to patients from other countries [32]. In this study, customized rTMS methods were applied to individual patients. This was a factor that influenced our results since the effects of rTMS depend on the irradiation method implemented. In addition, occupational therapy in NEURO was provided by therapists affiliated with NEURO-accredited facilities based on a pre-defined concept. Therefore, it was expected that treatment equivalence had been ensured; however, we were unable to confirm this point as part of this study. The generalization of the results is limited because the treatment effect will not be the same if the therapy is provided outside of a NEURO-accredited facility. The content and number of exercises provided to patients influence motor paralysis recovery [54,55]. Since the patients in the present analysis received NEURO at an accredited facility because they had chosen to undergo NEURO, it was inferred that they were highly motivated to undergo rehabilitation. Patient motivation and effort during inpatient treatment are confounding factors that influence treatment efficacy [75]. In the present study, we could not investigate the specific content and amount of exercise provided by the therapists nor the effort capacity of the patients. Physiotherapy sessions were assigned at the discretion of the attending physicians, but we have not been able to verify whether the sessions had an influence on the treatment effect of upper limb motor function. In the present study, patients for whom at least 6 months had elapsed since the onset of illness, which is the criterion for application of NEURO, were included in the study. In the acute phase, ipsilesional facilitation is performed by HF-rTMS [76]. In the chronic phase, rTMS was applied for interhemispheric inhibition, and the irradiation method differed from that in the acute phase according to the purpose of neuromodulation. The effectiveness of treatment for patients in the early stages of disease onset should be verified by other studies. In addition, in the present study, brain imaging data were not examined in detail. Given that some previous studies have shown that stroke subtype is a confounding factor for recovery, a detailed analysis of the neurological characteristics of patients receiving NEURO should be conducted to fully understand this issue [77].

## 5. Conclusions

This study estimated the level of hand and upper limb function restoration resulting from NEURO treatment, according to the severity of motor paralysis assessed in pre-treatment ARAT scores. The results of the present study can be used to suggest patient-desired ADL exercises in occupational therapy after rTMS in accordance with the functional recovery of the paralyzed upper limb. The benefits of the paralyzed upper limb functional recovery were estimated, and the ADL exercises appropriate for functional recovery need to be verified in future studies. The results of this study may be applied in rehabilitation therapy using BCI, which has been developed in recent years, to set motor tasks according to patients’ motor paralysis. We expect that providing patients with motor tasks based on pre-intervention severity to predict post-treatment recovery will only be used in newly developed therapies in the future. 

## Figures and Tables

**Figure 1 brainsci-13-00284-f001:**
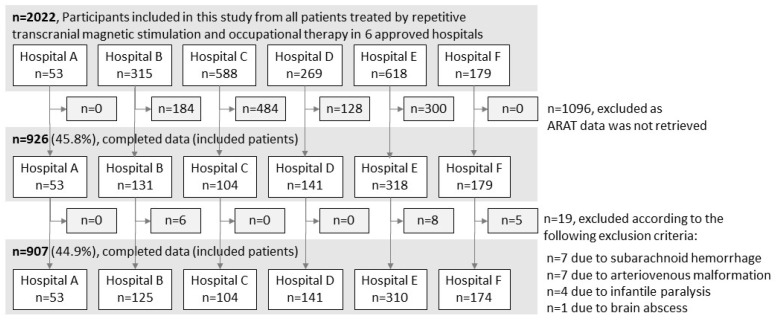
Patient selection procedure. ARAT, Action Research Arm Test.

**Figure 2 brainsci-13-00284-f002:**
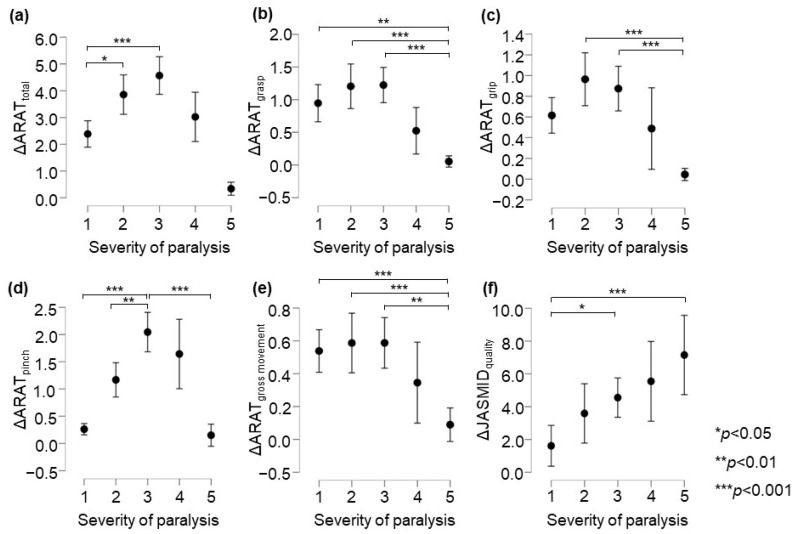
Comparison of changes in upper extremity function according to ARAT and JASMID scores. Panels are displayed delta score (post − pre) by severity of paralysis; (**a**) ARAT total, (**b**) ARAT grasp, (**c**) ARAT grip, (**d**) ARAT pinch, (**e**) ARAT gross movement, (**f**) JASMID quality. Statistical significance was set at *p <* 0.05 for Scheffé’s multiple comparisons (n = 907). ARAT, Action Research Arm Test; JASMID, Jikei Assessment Scale for Motor Impairment in Daily Living.

**Figure 3 brainsci-13-00284-f003:**
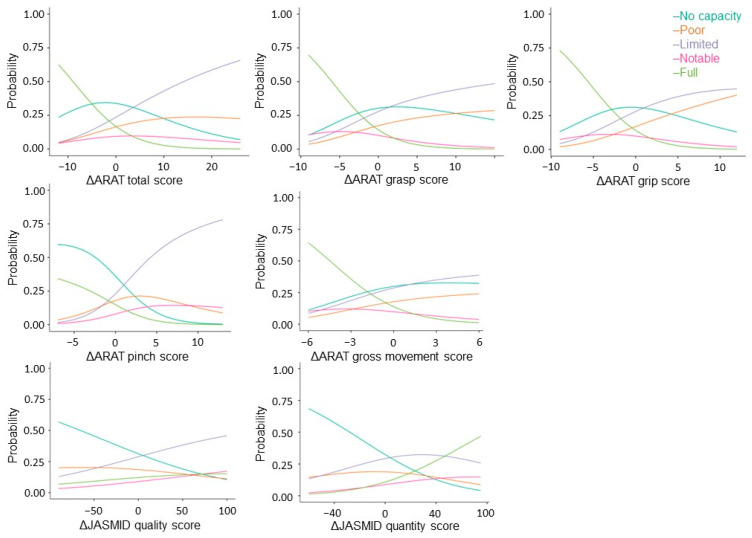
Multinomial logistic probability plots showing the association between the level of agreement for delta ARAT and JASMID score. Logistic curves were discriminated using the probability of being grouped by pre-treatment ARAT scores: scores of 0–10 indicate no upper limb capacity; scores of 11–21 represent poor capacity; scores of 22–42 represent limited capacity; scores of 43–54 represent notable capacity; and scores of 55–67 represent full upper limb capacity. ARAT, Action Research Arm Test; JASMID, Jikei Assessment Scale for Motor Impairment in Daily Living.

**Table 1 brainsci-13-00284-t001:** Clinical characteristics of analyzed patients.

Characteristics		All (n = 907)
Age (years)		63 (53, 70)
Sex	Female	297 (33)
	Male	610 (67)
Paralyzed hand	Left	395 (44)
	Right	512 (56)
Dominant hand	Left	44 (5)
	Right	861 (95)
Laterality in the paretic and dominant hand	Bilateral	2 (0.2)
	Ipsilateral side	411 (45)
	Contralateral side	496 (54)
Diagnosis	CI	465 (51)
	ICH	442 (49)
Time from onsets (months)		40 (22, 68)
rTMS stimulation method	Low frequency	708 (78)
	High frequency	1 (0.1)
	Theta burst	198 (22)
Treatment by botulinum toxin A or xylocaine	Treatment	900 (99)
	No treatment	7 (0.8)
MAS of elbow flexor muscles	Grade 0	174 (19)
	Grade 1	288 (32)
	Grade 1+	254 (28)
	Grade 2	124 (14)
	Grade 3	13 (1)
	Grade 4	2 (0)
	Missing data	52 (6)
Pre-treatment ARAT score	Total	22 (8, 38)
Pre-treatment JASMID score	Quantity	34 (20, 57)
	Quality	31 (20, 50)

Values are presented as n (%) or median (25th, 75th percentile). CI, cerebral infarction; ICH, intracranial hemorrhage; rTMS, repetitive transcranial magnetic stimulation; MAS, Modified Ashworth Scale; ARAT, Action Research Arm Test; JASMID, Jikei Assessment Scale for Motor Impairment in Daily Living. Repetitive transcranial magnetic stimulation was set at 1 Hz for low-frequency stimulation, 10 Hz for high-frequency stimulation, and theta-burst as an alternative method. Total patients, n = 907.

**Table 2 brainsci-13-00284-t002:** Characteristics of analyzed patients according to ARAT severity classification.

Characteristics		Recovery Capacity on the ARAT				
		No	Poor	Limited	Notable	Full
Patients (n)		275 (30)	167 (18)	269 (30)	84 (9)	112 (12)
Age (years)		63 (53, 70)	66 (54, 71)	61 (52, 69)	64 (55, 70)	63 (54, 69)
Sex	Female	109 (40)	58 (35)	75 (28)	29 (35)	26 (23)
	Male	166 (60)	109 (65)	194 (72)	55 (65)	86 (77)
Paralyzed hand	Left	129 (47)	74 (44)	108 (40)	39 (46)	45 (40)
	Right	146 (53)	93 (56)	161 (60)	45 (54)	67 (60)
Dominant hand	Left	19 (7)	3 (2)	9 (3)	5 (6)	5 (6)
	Right	256 (93)	163 (98)	259 (96)	79 (94)	79 (94)
	Bilateral	0 (0)	1 (1)	1 (0.4)	0 (0)	0 (0)
Laterality in the paretic and dominant hand	Ipsilateral side	136 (49)	76 (46)	112 (42)	40 (48)	47 (42)
	Contralateral side	139 (51)	91 (54)	157 (58)	44 (52)	65 (58)
Diagnosis	CI	145 (53)	89 (53)	138 (51)	47 (56)	46 (41)
	ICH	130 (47)	78 (47)	131 (49)	37 (44)	66 (59)
Time from onset (months)		45 (27, 70)	40 (24, 63)	34 (19, 66)	37 (21, 78)	39 (18, 63)
rTMS stimulation method	Low frequency	222 (81)	124 (74)	206 (77)	67 (80)	89 (79)
	High frequency	1 (0.4)	0 (0)	0 (0)	0 (0)	0 (0)
	Theta burst	52 (19)	43 (26)	63 (23)	17 (20)	23 (20)
Treatment by botulinum toxin A or xylocaine	Treatment	2 (0.7)	1 (0.6)	2 (0.7)	1 (1)	1 (1)
	No treatment	273 (99)	166 (99)	267 (99)	83 (99)	111 (99)
Pre-treatment ARAT score	Total	4 (3, 6)	16 (13, 18)	30 (26, 36)	48 (45, 51)	57 (56, 57)
Pre-treatment JASMID score	Quantity	20 (19, 26)	28 (20, 38)	41 (28, 58)	60 (40, 77)	73 (59, 91)
	Quality	20 (20, 25)	26 (20, 35)	25 (37, 51)	48 (36, 63)	63 (48, 78)

Values are presented as n (%) or median (25th, 75th percentile). ARAT, Action Research Arm Test; CI, cerebral infarction; ICH, intracranial hemorrhage; rTMS, repetitive transcranial magnetic stimulation; JASMID, Jikei Assessment Scale for Motor Impairment in Daily Living. Repetitive transcranial magnetic stimulation was set at 1 Hz for low-frequency stimulation, 10 Hz for high-frequency stimulation, and theta-burst as an alternative method. Pre-treatment ARAT scores of 0–10 indicate no upper limb capacity, scores of 11–21 represent poor capacity, scores of 22–42 represent limited capacity, scores of 43–54 represent notable capacity, and scores of 55–67 represent full upper limb capacity. Total patients, n = 907.

**Table 3 brainsci-13-00284-t003:** Outcome scores according to ARAT severity classification.

Index of Measurements		ARAT Severity Classification				
		No	Poor	Limited	Notable	Full
Δ ARAT	Total	2.4 ± 4.2	3.9 ± 4.8	4.6 ± 5.8	3.0 ± 4.2	0.3 ± 1.3
	A. grasp	0.9 ± 2.4	1.2 ± 2.2	1.2 ± 2.2	0.5 ± 1.6	0.0 ± 0.5
	B. grip	0.6 ± 1.4	1.0 ± 1.7	0.9 ± 1.8	0.5 ± 1.8	0.0 ± 0.3
	C. pinch	0.3 ± 0.9	1.2 ± 2.1	2.0 ± 3.0	1.6 ± 2.9	0.2 ± 1.1
	D. gross movement	0.5 ± 1.1	0.6 ± 1.2	0.6 ± 1.3	0.3 ± 1.1	0.1 ± 0.5
Δ JASMID	Quantity	2.2 ± 10.9	3.3 ± 14.7	4.6 ± 12.0	5.1 ± 15.6	4.4 ± 15.5
	Quality	1.6 ± 10.5	3.6 ± 11.8	4.5 ± 10.0	5.5 ± 11.2	7.2 ± 12.9

Values are presented as mean ± standard deviation. ARAT, Action Research Arm Test; JASMID, Jikei Assessment Scale for Motor Impairment in Daily Living. Pre-treatment ARAT scores of 0–10, 11–21, 22–42, 43–54, and 55–57 represented no, poor, limited, notable, and full recovery capacity, respectively. Total patients, n = 907.

**Table 4 brainsci-13-00284-t004:** Differences between pre- and post-treatment ARAT and JASMID scores using analysis of covariance.

Index of Measurements		Pre-Treatment	Post- Treatment	Delta Value	F	*p*	η^2^
ARAT	Total	24.7 ± 18.5	27.9 ± 18.8	3.1 ± 4.8	18.68	<0.001	0.077
	A. grasp	8.4 ± 6.5	9.3 ± 6.5	0.9 ± 2.1	7.48	<0.001	0.032
	B. grip	5.4 ± 4.2	6.1 ± 4.3	0.7 ± 1.6	7.27	<0.001	0.031
	C. pinch	5.5 ± 6.4	6.5 ± 6.6	1.1 ± 2.3	29.41	<0.001	0.116
	D. gross movement	5.4 ± 2.7	5.9 ± 2.7	0.5 ± 1.1	4.82	<0.001	0.021
JASMID	Quantity	39.9 ± 25.2	44.0 ± 26.9	3.7 ± 13.1	2.02	0.089	0.009
	Quality	35.7 ± 21.1	40.1 ± 23.1	3.9 ± 11.1	6.66	<0.001	0.028

Analysis of covariance was used. Statistical significance was set at 0.05. Total patients, n = 907. Values are presented as mean ± standard deviation. ARAT, Action Research Arm Test; JASMID, Jikei Assessment Scale for Motor Impairment in Daily Living.

## Data Availability

The raw data supporting the conclusions of this article will be made available by the authors without undue reservation.

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
