# Peer review of "Upper Limb Function Recovery by Combined Repetitive Transcranial Magnetic Stimulation and Occupational Therapy in Patients with Chronic Stroke According to Paralysis Severity"

_brainsci, 2023, doi:10.3390/brainsci13020284_

Round 1

Reviewer 1 Report

Undoubtedly, target activities of daily living using upper limb functions can be established by predicting the amount of change after treatment for each paralysis severity level to further aid practice planning. Authors in this paper estimated post-treatment score changes for each severity level of motor paralysis (no, poor, limited, notable, and full), stratified according to Action Research Arm Test scores before combined rTMS and intensive occupational therapy. 

My comments on the article are as follows:

- I propose to extend the introduction with a reference to training in the field of motor imagery using BCI technology. For example, you can refer to the publication: "Brain-computer technology based training system in the field of motor imagery", IET Science Measurement and Technology, WILEY, 2020.

- I suggest expanding the article with more data/graph simulations. The article presents a lot of data in tabular form, but few figures, which negatively affects cognitive values.

- Conclusion is too short. They should be extended, inter alia, with plans for the future in the scope of the proposed research work.

- The article lacks a bibliography, although citations are included in the text. This should be extended.

- Take care of the general editing aspect of the article.

Author Response

Reply

Reviewer #1

Comments and Suggestions for Authors

Undoubtedly, target activities of daily living using upper limb functions can be established by predicting the amount of change after treatment for each paralysis severity level to further aid practice planning. Authors in this paper estimated post-treatment score changes for each severity level of motor paralysis (no, poor, limited, notable, and full), stratified according to Action Research Arm Test scores before combined rTMS and intensive occupational therapy.

My comments on the article are as follows:

Comment 1

- I propose to extend the introduction with a reference to training in the field of motor imagery using BCI technology. For example, you can refer to the publication: "Brain-computer technology based training system in the field of motor imagery", IET Science Measurement and Technology, WILEY, 2020.

Response to Comment 1

Thank you for your suggestion. We speculate that the results of this study may be applied for the development of therapies using brain-computer interfaces. We have added this point to the introduction and discussion.

“Recently, a treatment method using brain-computer interface (BCI) was developed for the rehabilitation of stroke patients, and its effectiveness has been reported [21,22]. Even for new intervention methods, it is better to formulate exercises adapted to the severity of paralysis and recovery. Therefore, the results obtained in this study can be used as data to plan the most appropriate practice for patients in terms of future new intervention methods.” (1. Introduction, lines 80–85)

“Reference #21 Paszkiel, S.; Dobrakowski, P. Brain–computer technology-based training system in the field of motor imagery. IET Science, Measurement & Technology. 2020, 14, 1014–1018.”

“Reference #22 Baniqued, P.D.E.; Stanyer, E.C.; Awais, M.; Alazmani, A.; Jackson, A.E.; Mon-Williams, M.A.; Mushtaq, F.; Holt, R.J. Brain-computer interface robotics for hand rehabilitation after stroke: a systematic review. J Neuroeng Rehabil. 2021, 18, 15.”

“The results of this study can be applied to new interventions, including BCI, when planning exercises appropriate to the severity of the patient's condition. For example, a treatment in which BCI was applied to exercise therapy provided by therapists was reported to result in greater improvement in patients' motor paralysis [67,68]. Although the therapeutic effects of BCI have been demonstrated, a method to predict patient recovery and to set motor tasks appropriate for their severity has not been formulated [69]. The results of this study provided data to predict the amount of recovery after treatment for patients to whom newly developed interventions will be applied and to plan effective exercise tasks according to the severity of the patient's illness.” (4. Discussion, lines 425–433)

“Reference #67 Zhao, C.G.; Ju, F.; Sun, W.; Jiang, S.; Xi, X.; Wang, H.; Sun, X.L.; Li, M.; Xie, J.; Zhang, K.; et al. Effects of Training with a Brain-Computer Interface-Controlled Robot on Rehabilitation Outcome in Patients with Subacute Stroke: A Randomized Controlled Trial. Neurol Ther. 2022, 11, 679–695.”

“Reference #68 Prasad, G.; Herman, P.; Coyle, D.; McDonough, S.; Crosbie, J. Applying a brain-computer interface to support motor imagery practice in people with stroke for upper limb recovery: a feasibility study. J Neuroeng Rehabil. 2010, 7, 60.”

“Reference #69 Fu, J.; Chen, S.; Jia, J. Sensorimotor Rhythm-Based Brain-Computer Interfaces for Motor Tasks Used in Hand Upper Extremity Rehabilitation after Stroke: A Systematic Review. Brain Sci. 2022, 13.”

Comment 2

- I suggest expanding the article with more data/graph simulations. The article presents a lot of data in tabular form, but few figures, which negatively affects cognitive values.

Response to Comment 2

Thank you for your suggestion, and we have accordingly added Figure 3 (3.4. Main Results, lines 313–317). Table 5 is attached as Appendix 1 because the data to be used for predicting recovery from motor paralysis. We have added this point to the Statistical Analysis and Main Results.

“To estimate post-treatment recovery from pre-treatment paralysis severity, a multinomial logistic regression analysis was performed using delta values of ARAT and JASMID as dependent variables and pre-treatment motor paralysis severity as a predictor. ” (2.7. Statistical Analysis, lines 202–205)

“Next, stratified analysis was performed using multinomial regression to factor changes in ARAT and JASMID scores into the severity of motor paralysis prior to treatment. Akaike’s Information Criterion (AIC)=2677, grasp (x2=36.2, p<0.001, AIC=2724), grip (x2=31.9, p<0.001 , AIC=2728), pinch (x2=124, p<0.001, AIC=2635), gross movement (x2=21.1, p<0.001 , AIC=2739), and JASMID quality (x2=24.8, p<0.001, AIC=2735) showed a significant fit to the model by the severity of motor paralysis, but none in JASMID quantity (x2=6.52, p=0.16, AIC=2753). Coefficients of variation and odds ratios were calculated for the ARAT total score, subscores A–D, and the change in JASMID quantity and quality scores with respect to the No recovery group data for each severity level (Figure 3).” (3.4. Main Results, lines 290–298)

Comment 3

- Conclusion is too short. They should be extended, inter alia, with plans for the future in the scope of the proposed research work.

Response to Comment 3

We have added an addendum to the conclusion discussing the extension of the results of this study to rehabilitation therapy.

“The results of this study may be applied in rehabilitation therapy using BCI, which has been developed in recent years, to set motor tasks according to patients' motor paralysis. We expect that providing patients with motor tasks based on pre-intervention severity to predict post-treatment recovery will only be used in newly developed therapies in the future.” (5. Conclusion, lines 475–479)

Comment 4

- The article lacks a bibliography, although citations are included in the text. This should be extended.

Response to Comment 4

Thank you for pointing this out. Accordingly, we have noted the treatment used in conjunction with occupational therapy.

“Treatments used in conjunction with occupational therapy included practice with visual stimulation of mirror images, transcutaneous electrical nerve stimulation, repetitive peripheral sensory stimulation, and muscle tendon vibration [25-28].” (2.5. rTMS Combined with Occupational Therapy, lines 152–155)

“Reference #25 Thieme, H.; Morkisch, N.; Mehrholz, J.; Pohl, M.; Behrens, J.; Borgetto, B.; Dohle, C. Mirror therapy for improving motor function after stroke. Cochrane Database Syst Rev. 2018, 7, CD008449.”

“Reference #26 Marcolino, M.A.Z.; Hauck, M.; Stein, C.; Schardong, J.; Pagnussat, A.S.; Plentz, R.D.M. Effects of transcutaneous electrical nerve stimulation alone or as additional therapy on chronic post-stroke spasticity: systematic review and meta-analysis of randomized controlled trials. Disabil Rehabil. 2020, 42, 623–635.”

“Reference #27 Conforto, A.B.; Dos Anjos, S.M.; Bernardo, W.M.; Silva, A.A.D.; Conti, J.; Machado, A.G.; Cohen, L.G. Repetitive Peripheral Sensory Stimulation and Upper Limb Performance in Stroke: A Systematic Review and Meta-analysis. Neurorehabil Neural Repair. 2018, 32, 863–871.”

“Reference #28 Costantino, C.; Galuppo, L.; Romiti, D. Short-term effect of local muscle vibration treatment versus sham therapy on upper limb in chronic post-stroke patients: a randomized controlled trial. Eur J Phys Rehabil Med. 2017, 53, 32–40.”

Comment 5

- Take care of the general editing aspect of the article.

Response to Comment 5

The manuscript has been proofread by Editage ≺https://www.editage.jp/>. The editing certificate is attached to the cover letter.

We thank Reviewer #1 for their kind evaluation of the manuscript and for providing encouraging comments. We have revised the manuscript in accordance with the comments and concerns of Reviewer #1.

Reviewer 2 Report

We read with interest the study: "Upper limb function recovery by combined repetitive transcranial magnetic stimulation and occupational therapy in patients with chronic stroke according to paralysis severity."

The idea (research question) is interesting, and the study is overall well conceived. 

Abstract is concise, clear and complete.

Introduction is clear. 

M&M and results: nothing to concern. 

Discussion: I would suggest to add some minor comments regarding

- can this type of stimulation be applied even to recovery of the upper limb deficit in acute stroke? according to different NIHSS score (eg: Alexandre AM et al. Mechanical thrombectomy in acute ischemic stroke due to large vessel occlusion in the anterior circulation and low baseline National Institute of Health Stroke Scale score: a multicenter retrospective matched analysis. Neurol Sci. 2022 May;43(5):3105-3112. doi: 10.1007/s10072-021-05771-5. Epub 2021 Nov 29. PMID: 34843020.)

Conclusion: clear

Author Response

Reply

Reviewer #2

Comments and Suggestions for Authors

We read with interest the study: "Upper limb function recovery by combined repetitive transcranial magnetic stimulation and occupational therapy in patients with chronic stroke according to paralysis severity."

The idea (research question) is interesting, and the study is overall well conceived.

Abstract is concise, clear and complete.

Introduction is clear.

M&M and results: nothing to concern.

Conclusion: clear

Comment 1

Discussion: I would suggest to add some minor comments regarding

- can this type of stimulation be applied even to recovery of the upper limb deficit in acute stroke? according to different NIHSS score (eg: Alexandre AM et al. Mechanical thrombectomy in acute ischemic stroke due to large vessel occlusion in the anterior circulation and low baseline National Institute of Health Stroke Scale score: a multicenter retrospective matched analysis. Neurol Sci. 2022 May;43(5):3105-3112. doi: 10.1007/s10072-021-05771-5. Epub 2021 Nov 29. PMID: 34843020.)

Response to Comment 1

Thank you for your pertinent comment. The subjects included in this study were stroke patients 6 months after stroke onset; most of the selected rTMS irradiation methods included the low-frequency stimulation of the non-pathological cerebral hemisphere. rTMS for stroke patients in the early stage of stroke onset is reported to be effective when the diseased cerebral hemisphere is irradiated with high-frequency stimulation (Du J, 2019), because interhemispheric inhibition is related to the time of stroke onset, and patients in the early post-onset period have less imbalance in interhemispheric inhibition. rTMS has been shown to be effective for patients in the early post-onset period. However, rTMS guidelines warn that irradiation can induce seizures. Therefore, we suggest that the decision to use rTMS in stroke patients immediately after mechanical thrombectomy should be made by physicians with specialized knowledge and sufficient clinical experience.

We do not have data on acute-phase patients, because patients eligible for NEURO treatment are those for whom at least 6 months have elapsed since the onset of stroke. Because the effectiveness of rTMS treatment for early-stage stroke patients has been demonstrated (Komatsu, 2022), the effectiveness of the combination of rTMS and intensive occupational therapy for acute stroke patients needs to be examined separately, along with goal setting for activities of daily living. We have added this as a limitation of our study in the Discussion.

“In the present study, patients for whom at least 6 months had elapsed since the onset of illness, which is the criterion for application of NEURO, were included in the study. In the acute phase, ipsilesional facilitation is performed by HF-rTMS [76]. In the chronic phase, rTMS was applied for interhemispheric inhibition, and the irradiation method differed from that in the acute phase according to the purpose of neuromodulation. The effectiveness of treatment for patients in the early stages of disease onset should be verified by other studies.” (4. Discussion, lines 457–463)

“Reference #76 Komatsu, T.; Hada, T.; Sasaki, N.; Kida, H.; Takahashi, J.; Maku, T.; Nakada, R.; Shiraishi, T.; Akiyama, S.; Kitagawa, T.; et al. Effects and safety of high-frequency rTMS in acute intracerebral hemorrhage patients: A pilot study. J Neurol Sci. 2022, 443, 120473.”

The manuscript has been modified substantially in line with the reviewer’s suggestions, and we believe that the manuscript has benefitted from these revisions. Once again, we thank the reviewer for the time spent in reviewing the manuscript, and we look forward to meeting the reviewer’s expectations.

Round 2

Reviewer 1 Report

Dear Authors, 

Thank you for the changes made.

I accept the article in its current version.

Best regards, 

Szczepan Paszkiel